# AutoMoT: A Unified Vision-Language-Action Model with Asynchronous Mixture-of-Transformers for End-to-End Autonomous Driving

Wenhui Huang [* 1 2]   Songyan Zhang [* 1 3]   Qihang Huang [* 1]   Zhidong Wang [1]   Zhiqi Mao [1]   Collister Chua [1]
Zhan Chen [1]   Long Chen [3]   Chen Lv [† 1]

**AutoMoT Project Page**

## Abstract

Integrating vision-language models (VLMs) into end-to-end (E2E) autonomous driving (AD) systems has shown promise in improving scene understanding. However, existing integration strategies suffer from several limitations: they either struggle to resolve distribution misalignment between reasoning and action spaces, underexploit the general reasoning capabilities of pretrained VLMs, or incur substantial inference latency during action policy generation, which degrades driving performance. To address these challenges, we propose AutoMoT in this work, an end-to-end AD framework that unifies reasoning and action generation within a single vision-language-action (VLA) model. Our approach leverages a mixture-of-transformer (MoT) architecture with layer-wise joint attention sharing, which preserves the general reasoning capabilities of pre-trained VLMs while enabling efficient asynchronous inference over various tasks at different frequencies. Additionally, we explore a VLA-oriented action refiner that further enhances driving performance via diffusion-based fine-tuning. Extensive experiments on multiple benchmarks, under both open- and closed-loop settings, demonstrate that AutoMoT achieves state-of-the-art (SOTA) performance compared to existing methods. We further investigate the functional boundary of pre-trained VLMs in AD, examining when and to what extent AD-tailored fine-tuning is necessary.

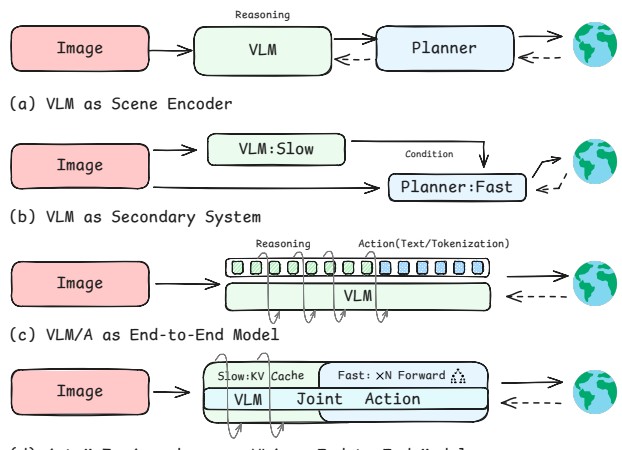

*Figure 1.* Comparison of different paradigms for integrating VLMs into conventional end-to-end autonomous driving frameworks. Our AutoMoT framework unifies reasoning and action-policy learning within a single vision–language–action (VLA) model through layer-wise joint attention sharing, while enabling asynchronous inference across tasks with different execution frequencies.

## 1. Introduction

The hierarchical modular pipeline, typically comprising perception, prediction, and planning, has been widely adopted in end-to-end (E2E) autonomous driving (AD) systems in recent years (Hu et al., 2023; Jiang et al., 2023; Liu et al., 2025a; Jaeger et al., 2023b; Liao et al., 2025). Recent advances in vision–language models (VLMs) have further benefited AD by enhancing high-level scene understanding, a capability that is often insufficient in conventional data-driven E2E systems when deployed in complex open-world scenarios. By leveraging their strong generalization and reasoning capabilities, VLMs endow AD systems with the potential to handle complex interactions and provide semantic explanations, thereby improving the interpretability.

The integration of vision–language models (VLMs) with end-to-end (E2E) autonomous driving systems is under-

---

[*]Equal contribution  [1]Nanyang Technological University, Singapore [2]Harvard University, US [3]Xiaomi EV, China. Correspondence to: Chen Lv <lyuchen@ntu.edu.sg>.

*Proceedings of the 43$^{rd}$ International Conference on Machine Learning*, Seoul, South Korea. PMLR 306, 2026. Copyright 2026 by the author(s).

going rapid development, giving rise to a diverse set of emerging design paradigms. A natural extension of the E2E framework incorporates VLMs into the upstream stages of the pipeline (Fu et al., 2025a; Li et al., 2025c), where pre-trained models provide rich scene understanding to support downstream planning, as illustrated in Fig. 1(a). Another line of work adopts a dual-system architecture (Fig. 1(b)), in which the VLM operates as an auxiliary module that assists conventional E2E pipelines by supplying high-level conditioning signals (Jiang et al., 2025; Tian et al., 2025). However, these approaches suffer from inherent distributional misalignment between the reasoning space of VLMs and the action space of planners. Furthermore, fine-tuning VLMs to generate intermediate conditioning signals inevitably constrains them to task-specific roles, diminishing the general capabilities of pretrained models.

More recently, as illustrated in Fig. 1c, emerging vision–language–action (VLA) architectures integrate reasoning and planning within a single pre-trained VLM backbone via autoregressive modeling (Wang et al., 2025; Zhou et al., 2026). While this unified design is compact and effectively leverages the strong reasoning capabilities of VLMs, tightly coupling action policy execution with high-level reasoning at a synchronized temporal frequency is impractical for real-world autonomous driving. This limitation becomes particularly severe in complex interactive environments, where low-latency control and rapid replanning are critical. Prior vision–language models that generate actions in textual form (Huang et al., 2026; Hwang et al., 2024; Zhang et al., 2024) can also be viewed as instances of this paradigm. In addition to the aforementioned limitations, these approaches rely on textual token supervision, which is inherently weaker than direct supervision on numerical action representations. Taking all these limitations into consideration, we pose the following key question: *How can VLA models fully leverage the generalist intelligence of a pre-trained VLM while mastering domain-specific capabilities and simultaneously satisfying real-time inference requirements?*

In this work, we propose **AutoMoT**, an end-to-end autonomous driving framework that seamlessly unifies asynchronous reasoning and action within a single vision–language–action (VLA) model, while avoiding both the degradation of VLM capabilities and distributional discrepancies across task spaces. As illustrated in Fig. 1d, AutoMoT adopts a mixture-of-transformers (MoT) architecture that bridges high-level reasoning (scene understanding) and low-level action policies (decision-making and trajectory planning) through joint attention in a shared latent space. This design enables asynchronous execution of textual reasoning and action generation at different temporal frequencies, thereby facilitating asynchronous inference. We comprehensively evaluate AutoMoT on both simulation

and real-world benchmarks under closed-loop and open-loop settings. Experimental results demonstrate competitive performance against state-of-the-art (SOTA) baselines, validating both the feasibility of the proposed framework and its effectiveness across diverse evaluation benchmarks. Moreover, through the comprehensive ablation studies, we found that pre-trained VLMs can achieve competitive multi-task scene understanding performance through semantic prompting alone, while fine-tuning remains essential for action-level tasks such as decision-making and trajectory planning.

The primary contributions of this work are as follows: 1) We propose AutoMoT, an end-to-end autonomous driving (AD) framework that seamlessly unifies scene understanding, decision-making, and planning within a single asynchronous VLA model via layer-wise joint attention sharing, while enabling asynchronous inference across tasks through different frequencies; 2) We investigate the functional boundaries of pretrained VLMs in autonomous driving, clarifying when and to what extent AD-specific fine-tuning is necessary across different tasks; 3) We explore a VLA-oriented action refiner that enhances driving performance through diffusion-based fine-tuning; 4) Extensive experiments demonstrate competitive performance against state-of-the-art baselines, validating both the feasibility of the proposed framework and its effectiveness across diverse evaluation benchmarks.

## 2. Related Work

### 2.1. End-to-End Autonomous Driving

Planning-oriented methods have been widely adopted in end-to-end autonomous driving frameworks in recent years. For instance, UniAD (Hu et al., 2023) proposes a hierarchical modular architecture that enables multiple tasks to be jointly learned in an end-to-end manner, mitigating error accumulation and consequently improving planning performance. The VAD series (Jiang et al., 2023; Chen et al., 2024) follows this design while introducing vectorized scene representations, which simplify the overall architecture and improve inference efficiency. Subsequently, Para-Drive (Weng et al., 2024) extends the hierarchical paradigm to a fully parallel formulation by unifying multiple tasks within the bird's-eye-view (BEV) space. More recently, diffusion-based policies (Chi et al., 2024) have attracted increasing attention in autonomous driving. Existing approaches typically apply diffusion models either as the core planner (Liao et al., 2025; Liu et al., 2025b) or as a trajectory refiner (Zhou et al., 2025a), leveraging their strong generative capabilities (Song et al., 2021; Ho et al., 2020) to improve driving performance. Nevertheless, these conventional end-to-end approaches still struggle with complex scene understanding, particularly when encountering long-tail and rare scenarios.

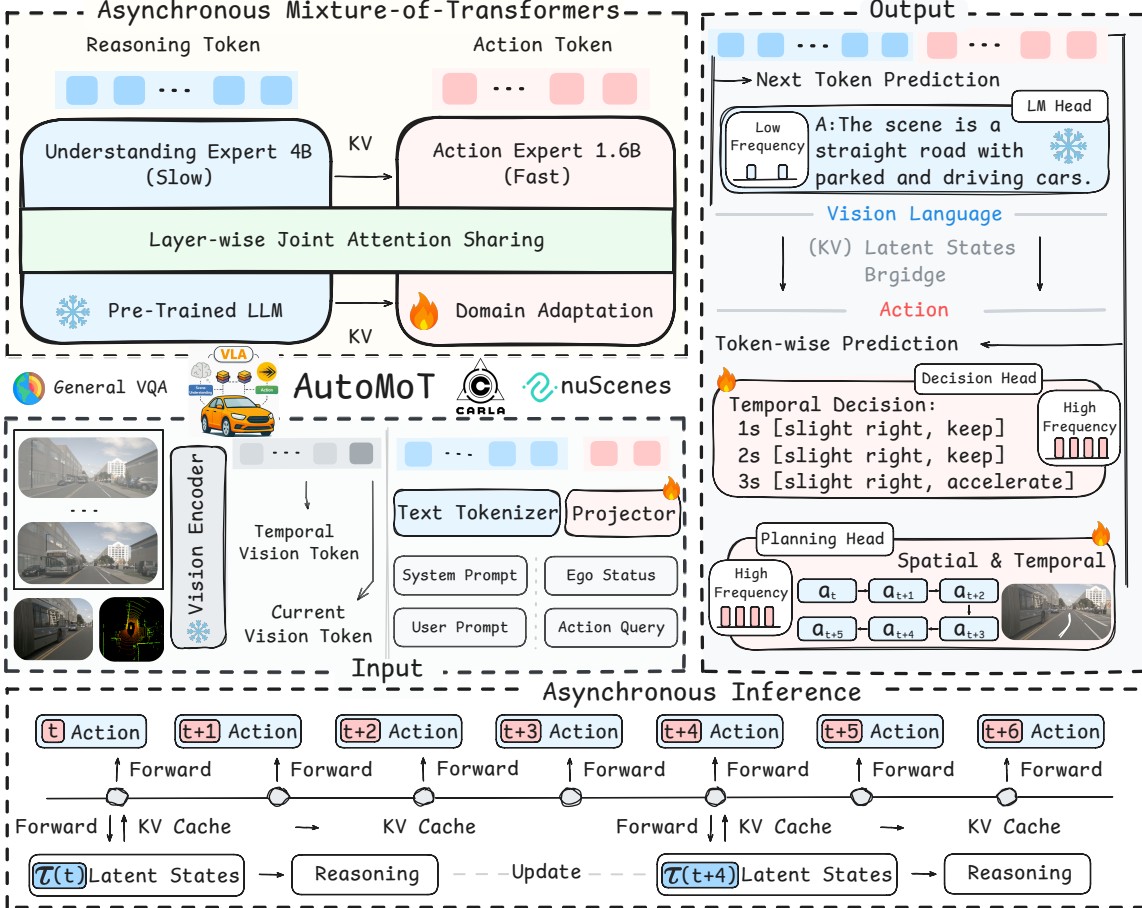

*Figure 2.* As an end-to-end autonomous driving framework, AutoMoT unifies scene understanding, decision-making, and trajectory planning within a single VLA model. AutoMoT adopts a MoT architecture that connects the understanding expert and the action expert via layer-wise joint attention sharing, while enabling asynchronous inference at different frequencies through KV Caching. A VLA-oriented action refiner is further explored to enhance driving performance via diffusion-based refinement.

## 2.2. Vision-Language Models for Autonomous Driving

The strong scene understanding and semantic reasoning capabilities of VLMs have motivated their rapid integration into E2E AD systems, resulting in several emerging design paradigms. Representative works such as Orion (Fu et al., 2025a) and ReCogDrive (Li et al., 2025c) introduce VLMs as upstream modules to enhance scene understanding and interpretability. Another line of work incorporates VLMs as secondary systems through intermediate representations, where DriveVLM (Tian et al., 2025) generates initial trajectory proposals, while Senna (Jiang et al., 2024) and ReCog-Drive (Li et al., 2025c) provide high-level decisions to guide downstream planning. Vision–language–action (VLA) architectures, including AutoVLA (Zhou et al., 2025b), Sim-lingo (Renz et al., 2025), and Alpamayo-R1 (Wang et al., 2025), further unify multiple tasks within a single pre-trained VLM backbone. However, the single-transformer design tightly couples reasoning and planning at a synchronized frequency, resulting in substantial inference latency, especially when chain-of-thought (CoT) reasoning is re-

quired for complex scene understanding. In contrast, our MoT-based VLA architecture systematically unifies scene understanding, decision-making, and planning in a single model through joint attention sharing (Deng et al., 2025; Huang et al., 2025a), while remaining functionally decomposed. This design enables asynchronous inference at different frequencies, thereby alleviating latency bottlenecks.

## 3. AutoMoT

### 3.1. Network Architecture

The overall framework of AutoMoT is illustrated in Fig. 2. AutoMoT comprises three core components: a scene understanding expert, an action expert, and an action refiner, all implemented using transformer-based architectures. In the following sections, we detail the design of each component as well as its corresponding training strategies. A detailed illustration of the data flow in AutoMoT is provided in Fig. 5 of Appendix A.1.

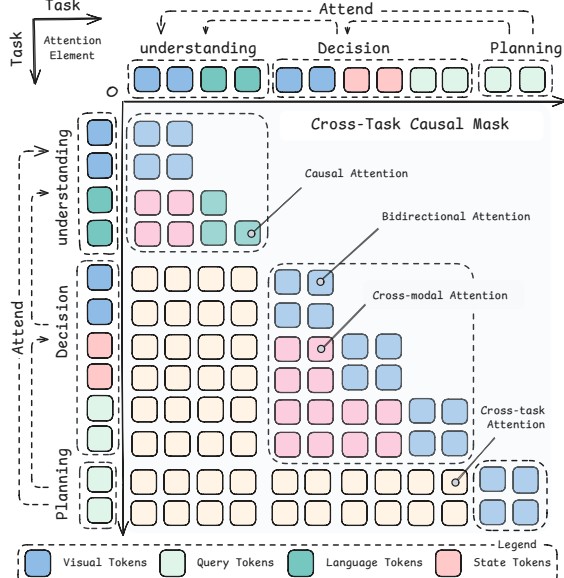

*Figure 3.* Our mask coordinates understanding, decision-making, and planning within a unified attention space. It enables intra-task multi-modal aggregation and cross-task information flow while preserving task-level causal ordering. This hybrid design maintains hierarchical causality and supports rich contextual integration, enabling AutoMoT to achieve coherent multi-task reasoning, decision-making, and trajectory planning.

**Understanding Expert**   The primary role of the understanding expert (UE) in AutoMoT is to perform scene understanding and generate chain-of-thought (CoT) reasoning for complex scenarios, particularly long-tail and rare cases, while transferring its general knowledge to facilitate action policy learning. The UE adopts Qwen3-VL-4B dense model as its vision–language backbone, which takes as input multi-view and multi-frame RGB images $I^{RGB} \in \mathbb{R}^{N \times H \times W \times C}$ captured by onboard cameras, together with textual prompts $\ell$ consisting of system prompts and user instructions, and outputs semantic reasoning results. To fully leverage the general knowledge of the pretrained Qwen3-VL model and avoid catastrophic degradation of reasoning performance, we freeze the understanding expert throughout the entire training process. The rationale behind this design is further investigated and discussed in Section 4.3.

**Action Expert**   The Action Expert (AE) in AutoMoT is responsible for decision-making and trajectory planning within the unified VLA framework. At each timestep $t$, the AE takes the current observation $o_t = \{I_t^{RGB}, I_t^{BEV}, Q(t)\}$ as input and produces action-side latent representations. Here, $I_t^{BEV}$ denotes the LiDAR BEV feature and $Q(t)$ represents the action queries. From these latent representations, the layer-wise query, key, and value embeddings $\{Q^l(t), \tilde{K}^l(t), \tilde{V}^l(t)\}$ are derived, where $l$ indexes the l-th attention layer. Based on these latent representations, the AE generates semantic decisions for the next three consecutive frames (semantic decisions for

the next 3 seconds), along with temporal and spatial trajectory proposals over the same horizon. More specifically, given the current observation $o_t$ and a set of action queries $Q(t)$, the AE jointly produces latent representations for decision-making and trajectory planning. These representations are decoded into three outputs: (i) concrete meta-actions $\hat{Z}_t = \{\hat{z}_{t+h}\}_{h=1}^H$, (ii) future temporal waypoints $\hat{Y}_t = \{\hat{y}_{t+m}\}_{m=1}^M$, and (iii) spatial route points $\bar{Y}_t = \{\bar{y}_{t+n}\}_{n=1}^N$. Here, $H = 3$ denotes a 3-second prediction horizon at 1s intervals for meta-actions, $M = 6$ denotes temporal waypoints sampled at 0.5s intervals over the same horizon, and $N$ represents the number of spatial route nodes used to parameterize the reference path. Notably, language, cross-modal, and cross-task interactions are constrained to follow causal attention, while intra-task and self-modal interactions adopt bidirectional attention.

By operating in a shared attention space with the UE, the AE conditions the latent reasoning generated by the UE into the action generation process, thereby grounding decision-making and planning in high-level scene understanding and enabling knowledge transfer from the pretrained VLM to policy learning. The attention patterns are visualized in Fig. 3. As shown, understanding, decision-making, and planning are regulated through cross-task causal attention, where decision representations are conditioned on understanding, and planning is further conditioned on both understanding and decision in the latent space. Within each task, latent features follow bidirectional attention across modalities, while cross-task interactions are governed by causal attention. The AE is implemented as a task-specialized transformer with approximately 1.6B parameters and is trained from scratch to capture domain-specific knowledge for autonomous driving. Notably, the AE operates at a higher frequency than the UE, enabling efficient inference and supporting real-time autonomous driving in complex environments.

**Action Refiner**   Recently, generative planners such as diffusion policies (Chi et al., 2024) have demonstrated strong potential for autonomous driving. In our framework, we implement the policy module as a diffusion-based policy built on the Diffusion Transformer (DiT). Instead of starting the reverse process from clustered trajectories (Zou et al., 2025) or pure white noise (Chi et al., 2024), we use the spatial-temporal trajectories predicted by the AE as informative priors and perform truncated reverse denoising to generate the refined trajectories. This design provides a more reliable initialization and significantly accelerates inference.

Concretely, the AE trajectory proposals are perturbed with multiplicative Gaussian noise (Zou et al., 2025), formulated as $\tau' = (1+\epsilon_{\text{mul}}) \odot \tau$. Based on the noisy trajectory samples, we construct temporal trajectory queries $Q_{\text{temp}} \in \mathbb{R}^{B \times M \times 2}$, and the spatial queries $Q_{\text{spatial}} \in \mathbb{R}^{B \times N \times 2}$, and concatenate them as: $X = [Q_{\text{temp}} \| Q_{\text{spatial}}]$, and processed by a stack of

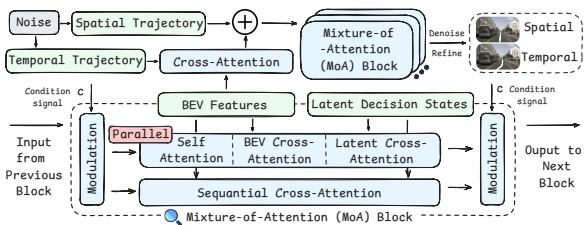

*Figure 4.* The architecture of DiT-based action refiner with Mixture-of-Attention blocks.

$N$ DiT decoder blocks. The conditioning signal $c$, which integrates the diffusion timestep, the current ego state, and lower-dimensional state history, is injected into each block through adaptive layer normalization (AdaLN).

To effectively exploit heterogeneous information during denoising, the diffusion policy leverages two complementary sources: the latent decision states $h_{\text{de}}$ from the AE for decision-aware trajectory generation, and the BEV feature $F_{\text{bev}}$ from the vision encoder for spatial guidance. Existing diffusion planners either flatten heterogeneous modalities into a single token sequence for joint attention (Li et al., 2025c), or process them through sequential cross-attention (Liao et al., 2025). The former dilutes the structural guidance of trajectory priors, while the latter imposes a fixed attention ordering that ties modality importance to processing sequence. To address this issue, we propose a Mixture-of-Attention (MoA) mechanism (Fig. 4) that adaptively balances attention sources between language and vision, enabling parallel and adaptive multi-source fusion while preserving anchor trajectory priors.

Specifically, MoA adopts a main-bypass fusion design. In the main pathway, attention is computed in parallel over three sources: self-attention among temporal and spatial queries, cross-attention to BEV features, and cross-attention to latent decision states. In addition, the contribution of latent decision states is modulated by a learnable factor $g = \tanh(\gamma)$, enabling adaptive control over multi-frame meta-actions. To further stabilize information propagation across diffusion stages, we introduce residual bypass pathways that preserve global contextual cues from different modalities. Specifically, $R_{\text{bev}}$ is obtained by mean pooling over BEV features, while $R_{\text{reason}}$ is obtained by attention pooling over reasoning tokens. The final fused representation is computed as $X' = X + \alpha \cdot (O_{\text{main}} + \sigma(\beta_b)R_{\text{bev}} + \sigma(\beta_r)R_{\text{reason}})$, where $O_{\text{main}}$ denotes the fused output of the main pathway, $\alpha$ is a scaling factor derived from AdaLN conditioned on $c$, and $\sigma(\beta_b)$ and $\sigma(\beta_r)$ are learnable gating coefficients.

### 3.2. Training Strategy

**Decision Making** We formulate decision-making as a token-level sequence modeling problem over meta-actions, conditioned on multi-frame driving observations. For real-world evaluation, we construct a multi-frame decision-making dataset based on nuScenes, termed *NuSync*.

Specifically, *NuSync* takes four consecutive historical RGB observations along with an additional RGB-BEV pair as input. In the synchronous setting, the RGB-BEV pair shares the same timestamp as the last historical frame, i.e., $I_t^{\text{sync}} = \{I_t^{RGB}, I_{t+1}^{RGB}, I_{t+2}^{RGB}, I_{t+3}^{RGB}, I_{t+3}^{RGB}, I_{t+3}^{BEV}\}$. In the asynchronous setting, we construct temporally asynchronous samples in which the four historical frames remain consecutive, while RGB–BEV pairs are constructed at both 1-frame and 2-frame future offsets, corresponding to 0.5s and 1s at 2Hz. For example, $I_t^{\text{async}} = \{I_t^{RGB}, I_{t+1}^{RGB}, I_{t+2}^{RGB}, I_{t+3}^{RGB}, I_{t+k}^{RGB}, I_{t+k}^{BEV}\}$, where $k \in \{4, 5\}$. In the output space, *NuSync* annotates meta-actions over a 3-second horizon, providing up to twenty possible combinations of longitudinal and lateral actions at 1s, 2s, and 3s. After curation, *NuSync* contains 80.1K samples in total. More details about *NuSync* and the associated benchmark are provided in Appendix A.2. To the best of our knowledge, *NuSync* is the first open-source decision dataset that supports asynchronous multi-frame meta-action inference. Similarly, for CARLA simulation, we construct the *PDM-Meta* dataset based on PDM-Lite following the same protocol. Due to the ambiguous boundaries between lateral meta-actions in simulation, we only annotate longitudinal decisions.

Based on the constructed meta-action datasets, given an observation sequence $o_t$, the AE predicts a sequence of meta-action tokens $\hat{z}_t = \{\hat{z}_t^j\}_{j=1}^J$, where $j$ represents j-th token and M depicts the necessary amount of tokens to be encoded as a meta-action. Unlike the next-token prediction used by the UE, the AE adopts a token-wise prediction paradigm and optimizes the policy by minimizing the negative log-likelihood of the target decision tokens:

$$\mathcal{L}_{\text{DM}} = \mathbb{E}_{(o_t, z_t) \sim \mathcal{D}} \left[ -\sum_{j=1}^{J} \log p_\theta \left( z_t^j \mid o_t \right) \right]. \quad (1)$$

where $\mathcal{D}$ denotes the corresponding dataset.

**Trajectory Planning** AutoMoT follows the original nuScenes setting, using four historical frames to predict a 3-second future trajectory. Following PDM-Lite, the 3-second trajectory is represented by temporal waypoints for capturing motion dynamics and 20 spatial route points for road geometry. For the AE, we optimize the trajectory planning with an $\ell_1$ loss:

$$\mathcal{L}_{\text{traj}}^{\text{temp}} = \mathbb{E}_{(o_t, Y_t^{\text{temp}}) \sim \mathcal{D}} \left[ \frac{1}{M} \sum_{m=1}^{M} \left\| \hat{Y}_{t+m} - Y_{t+m}^{\text{temp}} \right\|_1 \right],$$

$$\mathcal{L}_{\text{traj}}^{\text{spatial}} = \mathbb{E}_{(o_t, Y_t^{\text{spatial}}) \sim \mathcal{D}} \left[ \frac{1}{N} \sum_{n=1}^{N} \left\| \bar{Y}_{t+n} - Y_{t+n}^{\text{spatial}} \right\|_1 \right],$$

$$(2)$$

*Table 1.* Comparison of closed-loop planning performance on the CARLA Bench2Drive leaderboard. DS and SR represent Driving Score and Success Rate, respectively.

| Method | Expert | VLM | Generative Planner | Closed-loop Metric | |
|---|---|---|---|---|---|
| | | | | DS↑ | SR(%)↑ |
| MomAD (Song et al., 2025) | Think2Drive | - | - | 44.54 | 16.71 |
| UniAD-Base (Hu et al., 2023) | Think2Drive | - | - | 45.81 | 16.36 |
| TCP-traj (Wu et al., 2022) | Think2Drive | - | - | 59.90 | 30.00 |
| DriveTransformer-Large (Jia et al., 2025) | Think2Drive | - | - | 63.46 | 35.01 |
| DriveAdapter (Jia et al., 2023) | Think2Drive | - | - | 64.22 | 33.08 |
| Raw2Drive (Yang et al., 2026) | Think2Drive | - | - | 71.36 | 50.24 |
| DiffusionDrive (Liao et al., 2025) | PDM-Lite | - | ✓ | 77.68 | 57.72 |
| TransFuser++ (Jaeger et al., 2023b) | PDM-Lite | - | - | 84.21 | 67.27 |
| ReasonPlan (Liu et al., 2025c) | Think2Drive | ✓ | - | 64.01 | 34.55 |
| Recogdrive (Li et al., 2025c) | Think2Drive | ✓ | ✓ | 71.36 | 45.45 |
| DriveMoE (Yang et al., 2025) | Think2Drive | ✓ | - | 74.22 | 48.64 |
| ORION (Fu et al., 2025a) | Think2Drive | ✓ | ✓ | 77.74 | 54.62 |
| SpaceDrive+ (Li et al., 2025a) | PDM-Lite | ✓ | - | 78.02 | 55.11 |
| MindDrive (Fu et al., 2025b) | Think2Drive | ✓ | ✓ | 78.04 | 55.09 |
| AutoVLA (Zhou et al., 2025b) | PDM-Lite | ✓ | - | 78.84 | 57.73 |
| SimLingo (Renz et al., 2025) | PDM-Lite | ✓ | - | 85.07 | 67.27 |
| AutoMoT (ours) | PDM-Lite | ✓ | - | **87.34** | **70.00** |
| AutoMoT + (ours) | PDM-Lite | ✓ | ✓ | **89.42** | **74.09** |

where $Y_t^{\text{temp}}$ and $Y_t^{\text{spatial}}$ denote ground-truth temporal and spatial trajectories. Notably, the decision-making and trajectory planning are jointly optimized within the AE, enabling AutoMoT to learn coherent action policies grounded in semantic representations from the UE.

To further enhance the driving behaviors, we employ a diffusion policy to refine the trajectory proposals. Unlike standard diffusion processes that initialize from Gaussian noise, we implement a truncated diffusion paradigm. This approach treats the proposal trajectories from the AutoMoT as an anchor, performing reverse denoising from this informed prior to the expert ground truth. For training and evaluation, we utilize the PDM-Lite dataset, which contains over 700,000 samples across 5,000+ scenarios. Each sample consists of 4 historical frames at a 2Hz sampling rate, with the model tasked to refine spatial-temporal waypoints proposed by AE. We employ L1 reconstruction error as the loss function for trajectory refinement.

### 3.3. Asynchronous Inference with KV Caching

We formulate asynchronous inference as a multi-rate process in which reasoning and action inference evolve at different temporal resolutions, while both remain grounded in real-time visual observations. The interaction between these two processes is mediated by a shared key–value (KV) cache, as illustrated in Fig. 2. At an arbitrary timestep $t$, given the current observation $o_t$, the AE derives layer-wise queries, keys, and values $\{Q_{\text{act}}^l(t), K_{\text{act}}^l(t), V_{\text{act}}^l(t)\}$ for each attention layer. Correspondingly, $\tau(t)$ denotes the time index of the most recent scene representation update available at action step $t$, satisfying $\tau(t) \leq t$. At the update time $\tau(t)$,

the UE produces a set of layer-wise KV representations, which are stored in a persistent KV cache:

$$\mathcal{C}^{\tau(t)} = \{K_{scene}^l(\tau(t)), V_{scene}^l(\tau(t))\}_{l=1}^L . \quad (3)$$

Therefore, the keys and values involved in the final attention computation are formed by combining the KV cache from the UE at time $\tau(t)$ with the KV representations derived from the AE at time $t$, which can be expressed as:

$$\begin{aligned} \tilde{K}^l(t) &= [K_{scene}^l(\tau(t)) \parallel K_{act}^l(t)], \\ \tilde{V}^l(t) &= [V_{scene}^l(\tau(t)) \parallel V_{act}^l(t)], \end{aligned} \quad (4)$$

where $[\cdot \| \cdot]$ denotes concatenation along the sequence dimension, and all keys and values share the same embedding dimensionality $d$. The joint attention is then computed as

$$\text{Attn}^l(t) = \text{softmax}\left(\frac{Q_{act}^l(t)\tilde{K}^l(t)^\top}{\sqrt{d}}\right)\tilde{V}^l(t) . \quad (5)$$

The joint attention and asynchronous inference with KV Caching constitute the core characteristics of AutoMoT. By allowing action inference to reuse scene representations that are updated at a different temporal frequency, the proposed framework enables decision-making and trajectory planning to operate with a higher execution frequency than scene understanding, while remaining grounded in real-time perceptual inputs. This design aligns with the real-time requirements of real-world autonomous driving.

*Table 2.* Comparison of the Open-loop planning in nuScenes. The ST-P3 evaluation protocol is used by default.

| Method | Ego Status | Finetuning | | | L2 (m) ↓ | | | | Collision (%) ↓ | | | |
| | | Understanding | Decision | Planning | 1s | 2s | 3s | Avg. | 1s | 2s | 3s | Avg. |
|---|---|---|---|---|---|---|---|---|---|---|---|---|
| UniAD (Hu et al., 2023) | Vector | - | - | ✓ | 0.44 | 0.67 | 0.96 | 0.69 | 0.04 | 0.08 | 0.23 | 0.12 |
| VAD (Jiang et al., 2023) | Vector | - | - | ✓ | 0.17 | 0.34 | 0.60 | 0.37 | 0.07 | 0.10 | 0.24 | 0.14 |
| Ego-MLP (Li et al., 2024) | Vector | - | - | ✓ | 0.15 | 0.32 | 0.59 | 0.35 | 0.00 | 0.27 | 0.85 | 0.37 |
| DriveTransformer-Large (Jia et al., 2025) | Vector | - | - | ✓ | 0.16 | 0.30 | 0.55 | 0.33 | 0.01 | 0.06 | 0.15 | 0.07 |
| AutoVLA (Zhou et al., 2025b) | Text | ✓ | - | ✓ | 0.21 | 0.38 | 0.60 | 0.40 | 0.13 | 0.18 | 0.28 | 0.20 |
| ORION(Chat-B2D)(Fu et al., 2025a) | - | ✓ | - | ✓ | 0.17 | 0.31 | 0.55 | 0.34 | 0.05 | 0.25 | 0.80 | 0.37 |
| RoboTron-Drive (Huang et al., 2025b) | - | ✓ | - | ✓ | 0.14 | 0.30 | 0.57 | 0.33 | 0.03 | 0.12 | 0.63 | 0.26 |
| OpenDrive-VLA (Zhou et al., 2026) | Text | ✓ | - | ✓ | 0.15 | 0.31 | 0.55 | 0.33 | 0.01 | 0.08 | 0.21 | 0.10 |
| OmniDrive (Wang et al., 2024) | Vector | ✓ | - | ✓ | 0.14 | 0.29 | 0.55 | 0.33 | 0.00 | 0.13 | 0.78 | 0.30 |
| EMMA[†] (Hwang et al., 2024) | Text | ✓ | - | ✓ | 0.14 | 0.29 | 0.54 | 0.32 | - | - | - | - |
| SpaceDrive (Zhou et al., 2025b) | Vector | ✓ | - | ✓ | 0.15 | 0.29 | 0.51 | 0.32 | 0.04 | 0.18 | 0.49 | 0.23 |
| OpenREAD (Zhang et al., 2025) | Vector | ✓ | - | ✓ | 0.17 | 0.34 | 0.56 | 0.36 | 0.04 | 0.08 | 0.22 | 0.11 |
| Drive-R1 (Li et al., 2026) | Text | ✓ | - | ✓ | 0.14 | **0.28** | 0.50 | **0.31** | 0.02 | 0.06 | 0.19 | 0.09 |
| DriveVLM-Dual (Tian et al., 2025) | Vector | ✓ | - | ✓ | 0.15 | 0.29 | **0.48** | **0.31** | 0.05 | 0.08 | 0.17 | 0.10 |
| OpenEMMA (Xing et al., 2025) | Text | - | - | - | 1.45 | 3.21 | 3.76 | 2.81 | - | - | - | - |
| AutoMoT (Ours) | Vector | - | ✓ | ✓ | **0.14** | 0.29 | 0.54 | 0.32 | **0.01** | **0.06** | **0.15** | **0.07** |

## 4. Experiments

### 4.1. Experimental Setup

**Datasets.** For the reasoning tasks, we evaluate the general performance of all models on both autonomous driving benchmarks and general-domain datasets, including OmniDrive (Wang et al., 2024), ScienceQA, and FigureQA. For action-level tasks, AutoMoT is primarily trained on three datasets: nuSync, which is annotated and curated in this work for decision-making, nuScenes (Caesar et al., 2020), and the CARLA-Garage dataset (Jaeger et al., 2023a) for trajectory planning. We follow the original training and evaluation protocols provided by the trajectory planning benchmarks. Additionally, as part of our ablation study, we fine-tune the understanding expert of AutoMoT exclusively on two autonomous driving VQA datasets, LingoQA (Marcu et al., 2024) and CODA-LM (Chen et al., 2025).

**Benchmarks and Metrics.** Scene understanding performance is evaluated on the LingoQA (Marcu et al., 2024) benchmark using its native metric, Lingo-Judge, as well as on other AD-tailored and general VQA datasets using GPT-based scores. We further evaluate the open-loop performance of AutoMoT on the nuScenes (Caesar et al., 2020) benchmark, using average accuracy (AA) for decision-making, as well as L2 distance and collision rate for trajectory planning. Closed-loop performance is assessed on the Bench2Drive (Jia et al., 2024) benchmark following the officially provided evaluation metrics.

**Implementation Details.** For action policy learning, we adopt a learning rate ranging from $1 \times 10^{-4}$ to $2 \times 10^{-5}$ and employ the Fully Sharded Data Parallel (FSDP) training strategy. All models are trained using eight NVIDIA A100 GPUs.

### 4.2. Main Results

In this section, we present detailed quantitative comparisons between AutoMoT and representative prior and SOTA methods across both reasoning and action-level tasks. Due to space limitations, detailed decision-making results are reported in Appendix A.2.

**Closed-Loop Planning Benchmark Results.** We first evaluate AutoMoT on a closed-loop evaluation benchmark and report the quantitative results in Table 1. It is clear that AutoMoT outperforms all VLM-augmented baseline methods and achieves SOTA performance in terms of both driving score (DS) and the success rate (SR). It is worth noting that SimLingo employs action dreamer–based simulation for data augmentation to increase the amount of training data, while AutoMoT is trained solely on the original dataset, yet dominates SimLingo (Renz et al., 2025) in terms of both main metrics. In addition, incorporating the AR further improves closed-loop driving performance, demonstrating the benefit of diffusion-based trajectory refinement. For AutoMoT +, we report the average performance over multiple runs, although higher peak performance is observed. This is because the intrinsic stochasticity of diffusion policies can be amplified in closed-loop driving, where small trajectory deviations may accumulate over time and lead to different outcomes.

**Open-Loop Planning Benchmark Results.** The open-loop planning performance of AutoMoT compared with various baselines is reported in Table 2. AutoMoT achieves competitive performance in terms of L2 displacement and

*Table 3.* Comparison of reasoning capabilities across both general-domain and autonomous driving–specific datasets. †: Results are reproduced using the official checkpoints and evaluation environments.

| Method | LingoQA | OmniDrive | CODA-LM | TallyQA | InfoVQA |
|---|---|---|---|---|---|
| ReCogDrive | **67.20** | 0.82 | 5.90 | 69.60 | 75.80 |
| Robotron-Drive† | 59.20 | 0.82 | **6.20** | 63.40 | 42.60 |
| OpenEMMA | 48.00 | 0.43 | 4.80 | 80.00 | 71.40 |
| AutoMoT | 67.00 | **0.89** | 6.07 | **81.40** | **89.30** |

attains SOTA results on collision rate. Notably, most existing methods adapt scene understanding to the autonomous driving domain by fine-tuning the VLM backbone, whereas only OpenEMMA and AutoMoT refrain from such domain-specific adaptation, yet exhibit a clear performance distinction in terms of L2 displacement. These results indicate that policy learning in VLA models plays a critical role in action-level tasks, extending beyond the inherent expertise of pre-trained VLMs. In contrast, fine-tuning the VLM backbone on AD datasets yields only marginal improvements in planning metrics. More importantly, such minor gains (e.g., a few centimeters in L2 displacement) may come at the cost of severe degradation in scene understanding capability due to catastrophic forgetting. To examine this trade-off more comprehensively, we further evaluate AutoMoT together with other open-source methods on both AD-specific and general-domain VQA datasets to assess their scene understanding performance.

**General VQA Benchmark Results.** Observations from the planning benchmarks further motivate a deeper investigation into whether additional domain-specific adaptation of scene understanding on top of pre-trained VLMs is indeed beneficial for overall autonomous driving performance. To clarify this question, we evaluate AutoMoT against other open-source baseline approaches on a diverse set of VQA benchmarks spanning both autonomous-driving–specific and general-domain tasks, as summarized in Table 3. Although both ReCogDrive and Robotron-Drive fine-tune their VLM backbones on LingoQA, OmniDrive, and CODA-LM, the resulting improvements are either marginal or even inferior to AutoMoT, which keeps the VLM backbone frozen. For example, ReCogDrive only marginally outperforms AutoMoT by 0.2 on the Lingo-Judge metric, while both ReCogDrive and Robotron-Drive underperform AutoMoT on the perception task of OmniDrive. More importantly, their performance on general-domain VQA benchmarks degrades substantially after fine-tuning, falling well below that of OpenEMMA and AutoMoT. Taken together with the analysis on planning benchmarks, these results suggest that fine-tuning the VLM backbone on AD-tailored scene understanding tasks provides only limited benefits for subsequent planning behaviors. Such gains are often marginal and accompanied by overfitting to specific benchmarks, leading to catastrophic forgetting and degraded generalization. In

*Table 4.* Ablation study results of investigating the performance boundary of the pre-trained backbone. †: System prompt is provided; ‡: Fine-tuned on autonomous driving datasets; L: Lingo-Judge; G:GPT-Score; A: Token Accuracy.

| Benchmark | Task Category | AutoMoT† | AutoMoT‡ |
|---|---|---|---|
| LingoQA (L) | Scene Understanding | 67.00 | **67.20** |
| OmniDrive (G) | Counterfactual Planning | 18.20 | **67.80** |
| ScienceQA (A) | General Knowledge | **88.60** | 87.80 |
| FigureQA (A) | General Knowledge | **97.60** | 91.20 |
| TallyQA (A) | General Knowledge | **81.40** | 52.40 |
| InfographicVQA (G) | General Knowledge | **89.30** | 50.20 |
| VizWiz (G) | General Knowledge | **75.60** | 50.20 |

the following section, we further investigate the functional boundaries of pre-trained VLMs in autonomous driving, examining whether domain-specific fine-tuning is necessary across different AD tasks.

### 4.3. Performance Boundary of Pretrained Backbone

In this section, we aim to investigate when and to what extent AD-tailored fine-tuning is beneficial under a more controlled and fair setting, as direct comparisons with existing methods are often confounded by various factors. To this end, we fine-tune the VLM backbone of AutoMoT on two autonomous driving datasets: LingoQA (Marcu et al., 2024) and the counterfactual reasoning subset of OmniDrive (Wang et al., 2024). The former is widely used for scene understanding in the autonomous driving domain, while the latter is closely related to planning performance, as it shares the same visual inputs as the nuScenes (Caesar et al., 2020) benchmark and its question prompts explicitly contain trajectory-related information. We then evaluate the fine-tuned model on the test splits of these two datasets, together with five additional general-domain knowledge benchmarks, ScienceQA (Lu et al., 2022), FigureQA (Kahou et al., 2017), TallyQA (Acharya et al., 2019), InfographicVQA (Mathew et al., 2022), and VizWiz (Gurari et al., 2018), as summarized in Table 4.

As shown by the quantitative results, fine-tuning the VLM backbone yields marginal improvements on scene understanding performance in LingoQA, but leads to substantial gains on the counterfactual planning task. This suggests that pre-trained VLMs can already support competitive multi-task scene understanding through semantic prompting alone, whereas fine-tuning remains essential for action-level tasks such as trajectory planning. Notably, the impact of fine-tuning on general-domain reasoning is highly task-dependent. On datasets with relatively simple answer spaces, such as ScienceQA and FigureQA, fine-tuning results in only minor performance changes, indicating that basic recognition and short-form reasoning capabilities are largely preserved. In contrast, for more complex VQA benchmarks that require compositional reasoning and multi-step inference, such as TallyQA, InfographicVQA, and VizWiz, per-

*Table 5.* Trajectory planning performance under synchronized and asynchronous settings. AutoMoT refers to the proposed model with the KV cache enabled, while AutoMoT-S denotes the synchronized variant that runs the understanding expert (UE) and action expert (AE) at the same frequency, without introducing temporal misalignment between UE and AE.

| Setting | L2@1s↓ | L2@2s↓ | L2@3s↓ | L2$_{avg}$ ↓ |
|---------|--------|--------|--------|--------------|
| AutoMoT-S | 0.140 | 0.290 | 0.537 | **0.322** |
| AutoMoT | 0.141 | 0.293 | 0.544 | 0.324 |

*Table 6.* Latency breakdown of AutoMoT under synchronized (AutoMoT-S) and default AutoMoT settings.

| Setting | Generative Planner | Time (ms) | | | | Frequency Hz |
|---------|---------|------|------|------|-------|-----------|
| | | UE | AE | AR | Total | |
| AutoMoT-S | - | 80.3 | 37.0 | - | 117.3 | 8.5 |
| AutoMoT | - | 0.0 | 37.0 | - | 37.0 | 27.0 |
| AutoMoT-S | ✓ | 80.3 | 37.0 | 26.0 | 143.3 | 7.0 |
| AutoMoT | ✓ | 0.0 | 37.0 | 26.0 | 63.0 | 16.0 |

formance degrades substantially after fine-tuning. For instance, accuracy on TallyQA drops from 81.40 to 52.40, and on InfographicVQA from 89.30 to 50.20, corresponding to an almost 50% reduction relative to the pre-trained baseline. These results demonstrate that domain-specific fine-tuning mainly degrades high-level reasoning ability, providing clear evidence of catastrophic forgetting when VLMs are directly adapted to the autonomous driving domain. A similar phenomenon has also been observed in other work (Li et al., 2025b).

Beyond validating our design choice, these results prompt a rethinking of the role of pre-trained VLMs in autonomous driving systems. Rather than uniformly adapting VLMs across all task levels, our findings suggest a clearer functional boundary: high-level scene understanding and reasoning do not necessarily require extensive domain-specific adaptation, whereas task-specific fine-tuning should primarily target action-level components that operate under domain-specific constraints. In this regard, our design preserves the general intelligence of the UE and transfers its reasoning knowledge to the AE through joint attention sharing, enabling effective action-level learning without sacrificing general reasoning ability.

### 4.4. Asynchronous versus Synchronous Inference

In this ablation, we study whether decoupling reasoning and action inference at different temporal resolutions hurts planning performance because the action expert may receive slightly outdated visual context. To test this, we build two dedicated validation settings by introducing controlled temporal offsets between the visual inputs used by the understanding expert (UE) and the action expert (AE). In the decoupled setting, UE and AE are asynchronous, with two-step (1.0 s) offsets. In the coupled setting, both experts operate synchronously. We compare AutoMoT, which uses asynchronous inference with a persistent KV cache, against a synchronized variant, AutoMoT-S, which disables the KV cache and recomputes both UE and AE outputs at every step.

Table 5 shows that AutoMoT retains nearly the same planning accuracy as the synchronized variant AutoMoT-S, despite the temporal decoupling between UE and AE. The

differences are small at all horizons: L2@1s changes from 0.140 to 0.141, L2@2s from 0.290 to 0.293, and L2@3s from 0.537 to 0.544. The average error increases only slightly, from 0.322 to 0.324, which corresponds to a relative degradation of 0.62%. This indicates that the stale visual context introduced by asynchronous execution has only a minor effect on planning quality.

At the same time, Table 6 shows a clear gain in efficiency. Without the diffusion head in the action expert, AutoMoT-S takes 117.3 ms per step, including 80.3 ms for preparing the KV cache in the understanding expert and 37.0 ms for the action expert, whereas AutoMoT reduces the total latency to 37.0 ms by reusing the cached understanding context, corresponding to a 68.5% reduction. For each asynchronous action step, the latency of the UE is reported as 0.0 ms because its representations are directly reused from the cached KV states. With the diffusion head enabled in the action expert, AutoMoT-S and AutoMoT achieve inference latencies of 143.3 ms and 63.0 ms, respectively. Overall, these results show that the proposed asynchronous inference scheme provides a substantial speedup while keeping planning performance almost unchanged. The inference latency comparison between AutoMoT and other VLA baselines is reported in Appendix A.4.

## 5. Conclusion

We propose AutoMoT, an asynchronous VLA model for end-to-end autonomous driving that unifies reasoning and action generation within a single MoT architecture via layer-wise joint attention sharing. AutoMoT preserves the general reasoning capability of the VLM backbone during action-policy learning, while enabling efficient asynchronous inference across tasks with different execution frequencies. Extensive evaluations on simulation and real-world benchmarks under both open- and closed-loop settings show that AutoMoT achieves SOTA performance against existing baselines, despite not fine-tuning the VLM backbone on AD-specific datasets. Moreover, experiments on both general-domain and AD-specific VQA benchmarks show that pre-trained VLMs already provide strong multi-task scene understanding through semantic prompting, whereas fine-tuning remains essential for action-level tasks in end-to-end autonomous driving.

## Acknowledgment

This work was supported by the Ministry of Education (MOE), Singapore, under Tier 2 Grant (MOE-T2EP50222-0002).

## Impact Statement

This work proposes a vision–language–action (VLA) model for end-to-end autonomous driving that unifies scene understanding, decision-making, and planning within a single framework, while enabling asynchronous inference across tasks with different frequencies and thus, better accommodating the real-time requirements of autonomous driving. While deploying VLA-scale models on current onboard computational platforms remains resource-intensive, rapid progress in computational hardware and model acceleration techniques, including quantization, pruning, and efficient caching, may make onboard deployment of Auto-MoT increasingly practical. In addition, we investigate the functional boundaries of pre-trained VLMs in autonomous driving, clarifying when and to what extent AD-specific fine-tuning is necessary for different task levels.

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

# A. Appendix for AutoMoT.

## A.1. Data Flow of AutoMoT

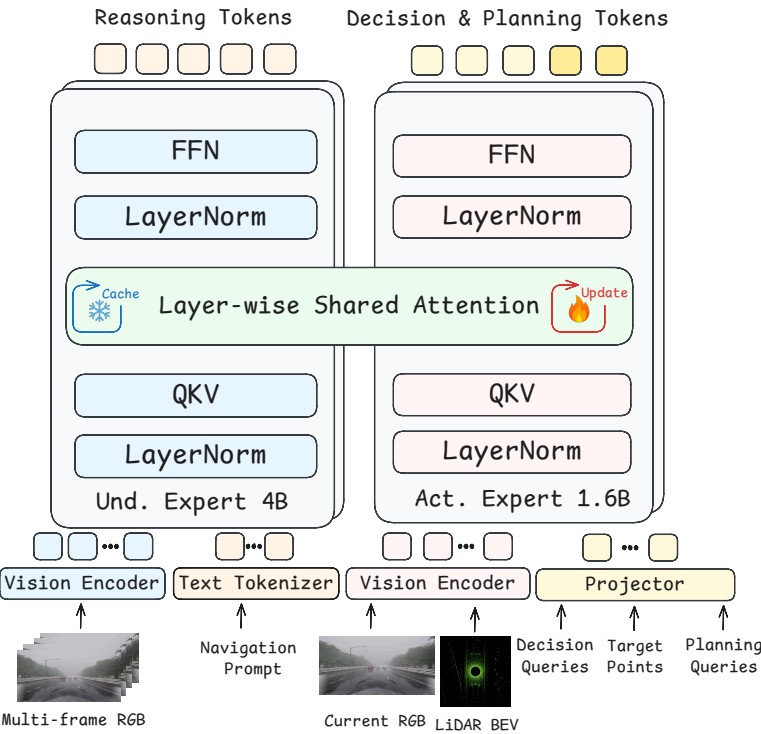

*Figure 5.* The detailed data flow of AutoMoT.

## A.2. Decision-Making Benchmark Results

**Decision Benchmark on NuSync Dataset.** In this section, we report the quantitative results of decision-making over the dataset constructed by ourselves, as mentioned in Section 3.2. The decision space consists of lateral actions, including *turn left*, *slight left*, *go straight*, *slight right*, *turn right*, and longitudinal actions, including *accelerate*, *slow*, *keep*, and *stop*, which contains consecutive meta-actions for three future time frames: 1s, 2s, 3s. Moreover, on top of this multi-frame decision-making dataset, we additionally construct an asynchronous version by decoupling the temporal alignment between semantic reasoning and action prediction. Specifically, the decoupled set contains asynchronous samples with one-step asynchrony (0.5 s) and two-step asynchrony (1.0 s). We then evaluate both AutoMoT (default asynchronous setup) and a coupled variant (AutoMoT-S) that disables the KV cache and runs UE and AE at the same frequency, recomputing all outputs at every step, and the results are shown in Table 7. We observe that although temporal asynchrony introduces additional uncertainty, the accuracy drop remains marginal, suggesting that KV-cache reuse effectively preserves semantic and temporal coherence across timesteps.

*Table 7.* Decision-making accuracy under synchronized and asynchronous settings at different time horizons.

| Method | Lateral Acc. ↑ | | | Longitudinal Acc. ↑ | | | Joint Acc. ↑ | | |
| --- | --- | --- | --- | --- | --- | --- | --- | --- | --- |
| | 1s | 2s | Avg | 1s | 2s | Avg | 1s | 2s | Avg |
| AutoMoT-S | 95.00% | 83.20% | 84.50% | 77.38% | 56.81% | 62.28% | 73.84% | 46.77% | 53.49% |
| AutoMoT | 94.06% | 82.69% | 83.79% | 77.57% | 56.85% | 62.38% | 73.40% | 46.36% | 53.10% |

**Decision Benchmark on Senna Dataset.** In order to confirm the superiority of AutoMoT, we further evaluate decision-making performance on the Senna meta-action benchmark constructed on nuScenes by following Senna (Jiang et al.,

2024). According to the original setting, Senna defines discrete meta-action labels by analyzing future trajectories, where longitudinal actions are categorized into *stop*, *accelerate*, *decelerate*, and *constant-speed* based on velocity variations, while lateral actions are categorized into *left/right turn*, *left/right lane change*, and *straight* based on lateral displacement and heading changes. These meta-action labels provide a structured abstraction of driving behaviors and enable systematic evaluation of high-level decision policies.

As shown in Table 8, AutoMoT achieves superior decision accuracy over the Senna baseline, demonstrating the superiority of our action expert in terms of action policy learning.

*Table 8.* Decision-making performance comparison on the Senna nuScenes benchmark.

| Model | Fine-tuned | Accuracy (%) |
|---|---|---|
| Senna | ✓ | 88.47 |
| AutoMoT (Ours) | ✓ | **90.92** |

### A.3. Impact of Individual Component

In this section, we conduct additional ablation studies to systematically analyze the contributions of individual components in AutoMoT, including the understanding expert, decision-making module (meta-action planning), and the asynchronous inference mechanism, with respect to overall driving performance. For consistency, all experiments are performed on the nuScenes benchmark, using the official trajectory planning metrics and average accuracy for decision-making evaluation.

*Table 9.* Ablation study on the understanding and decision-making components of AutoMoT. AutoMoT-R denotes the variant with a randomly initialized VLM backbone. AutoMoT-P denotes the planning-only variant, where the action expert is trained without the decision-making (meta-action planning) objective.

| Method | L2@1s $\downarrow$ | L2@2s $\downarrow$ | L2@3s $\downarrow$ | L2$_{avg}$ $\downarrow$ |
|---|---|---|---|---|
| AutoMoT (Ours) | 0.14 | 0.29 | 0.54 | 0.32 |
| AutoMoT-R (w/o Pre-trained UE) | 0.16 | 0.33 | 0.60 | 0.36 |
| AutoMoT-P (w/o decision-making) | 0.14 | 0.30 | 0.58 | 0.34 |

**Impact of the Understanding Expert.** To assess the importance of preserving the general intelligence of the VLM backbone, we replace the pre-trained Qwen3-VL-4B in the understanding expert with a randomly initialized counterpart and train it E2E on the trajectory planning task. As shown in Table 9, this from-scratch variant exhibits substantial performance drops across all planning horizons, indicating that the general-purpose knowledge and reasoning capabilities provided by the pre-trained understanding expert are crucial for stable and accurate planning. Notably, the degradation becomes more pronounced at longer horizons, suggesting that long-horizon trajectory planning relies heavily on high-quality scene understanding.

**Impact of the Decision-Making.** In this ablation study, we keep all components unchanged but remove the decision-making objective from the action expert (AE), training it solely on trajectory planning to quantify the contribution of decision-making to overall driving performance. Quantitative results are reported in Table 9. Removing decision-making consistently degrades performance across all prediction horizons, with the average L2 displacement increasing to 0.34.

### A.4. Inference Latency Comparisons

In this section, we compare AutoMoT against existing VLA baselines (Table 10) to further confirm the superiority of our method in terms of inference latency. Prior VLA-based systems operate at substantially lower frequencies, with reported inference latencies of 7683 ms for OpenEMMA, 1072 ms / 10518 ms for AutoVLA in fast / slow modes, and 430 ms for SimLingo. In contrast, AutoMoT achieves 117 ms under synchronous inference and 37 ms under asynchronous inference, corresponding to 8.5 Hz and 27 Hz, respectively. This yields an order-of-magnitude improvement in inference efficiency over existing VLA-based approaches. The gain is enabled by decoupling reasoning and action generation through asynchronous execution with KV-cache reuse: reasoning is updated at a lower frequency, while action generation runs at a higher frequency, avoiding redundant full-model inference. Compared with synchronized VLA formulations, AutoMoT substantially reduces

unnecessary computation and better aligns with the real-time requirements of autonomous driving, making VLA-based end-to-end systems more practical for onboard deployment.

*Table 10.* Inference latency comparison of VLA-based driving methods. †Results are cited from the original paper, as public checkpoints are not available for reproduction. The inference GPU is not specified; training uses $8\times$ NVIDIA L40S.

| Method | GPU | Latency (ms) | Hz |
|---|---|---|---|
| OpenEMMA | RTX 5090 | 7,683 | 0.13 |
| AutoVLA† (fast) | Unknown | 1,072 | 0.93 |
| AutoVLA† (slow) | Unknown | 10,518 | 0.10 |
| SimLingo | RTX 5090 | 430 | 2.3 |
| AutoMoT-S (sync) | RTX 5090 | 117 | 8.5 |
| AutoMoT (async) | RTX 5090 | **37** | **27** |

