# OpenReview forum: "AutoMoT: A Unified Vision-Language-Action Model with Asynchronous Mixture -of-Transformers for End-to-End Autonomous Driving"
_ICML.cc/2026/Conference — ICML 2026 regular_

### Official Review · Reviewer_Czzb · 2026-03-04

**Soundness:** 4
**Presentation:** 2
**Significance:** 3
**Originality:** 3
**Overall Recommendation:** 5
**Confidence:** 2

**Summary:**

This paper introduces AutoMoT, an end-to-end autonomous driving Vision-Language-Action (VLA) model framework that effectively integrates high-level scene reasoning with low-latency action planning within a single model.

**Compliance With Llm Reviewing Policy:**

Affirmed.

**Final Justification:**

The rebuttal fully resolved my specific concerns, so I am maintaining my positive score based on the paper's overall soundness and originality. However, as I acknowledge my limited familiarity with this specific subfield, I defer to the domain experts on the panel regarding the broader technical critiques.

**Key Questions For Authors:**

Q1: The claim on Page 7 (Lines 362-365) that the lower SR is primarily due to ‘AutoMoT performs a number of routes with a slight punishment, e.g., DS ≥ 95 with lane invasion, which ultimately cannot be attributed to the SR meaningfully’. Please supplement this with specific statistical data on the types of infractions or concrete case analyses to support this claim

**Limitations:**

L1: Deploying the model on onboard controllers or through vehicle-cloud collaboration will be hindered in practice by its massive scale and the transmission overhead of the KV cache.

**Strengths And Weaknesses:**

Strengths
S1: The paper achieves multi-dimensional innovations in its model architecture, effectively applying a Vision-Language-Action model to autonomous driving.
S2: The presentation is clear and complete.
S3: The paper provides a systematic and comprehensive experimental evaluation.
Weaknesses
W1: There is a spelling error in Figure 1(b): "Seconday System" should be corrected to "Secondary System".
W2: The representation of the "Input" module in Figure 2 is highly unclear. It fails to specify the data flow and final destination of the vision tokens, and it does not clearly illustrate which specific inputs the "Reasoning Token" and "Action Token" are derived from.

---

> ### Author Rebuttal · Authors · 2026-03-31
>
> We are deeply grateful to the reviewer for their insightful and thorough comments, and we appreciate the recognition of our work's contribution! Below, we provide point-by-point responses to all concerns (W).
>
> **W1: Typo in Figure 1(b).**
>
> We thank the reviewer for pointing this out. We will correct it as "Secondary System” and revise Figure 1(b) accordingly in the revised manuscript.
>
> ---
> **W2: Clarity of the Input module in Figure 2.**
>
> Thanks for pointing this issue out! We agree that Figure 2 lacks clarity in illustrating the data flow and token sources.
>
> To address this, we have added an additional diagram ([Figure Click Me!](https://tinyurl.com/mwf6awnr)) that explicitly presents the input space, MoT structure, and output space. We will include this new figure in the revised manuscript to provide a clearer illustration of the information flow and architecture.
>
> ---
> **W3: Analysis of SR degradation.**
>
> We thank the reviewer for this comment! We agree that the explanation of SR differences in the initial submission was not sufficiently supported. In earlier results, some failures corresponded to minor infractions (e.g., lane invasion) with high driving scores (DS ≥ 95), which led to lower SR.
>
> In the revised version, **this issue is largely mitigated** due to two refinements: (1) earlier injection of target-point information, which reduces shortcut behavior and improves alignment with scene context, and (2) joint training of spatial–temporal trajectory planning within the Action Expert, leading to more consistent action execution. As a result, AutoMoT now achieves **87.34 DS / 70.00% SR**, and such high-DS but failed cases are nearly eliminated. Across the 220 evaluation routes, we observe only a single case with DS = 99.02, and provide the demonstration in this [Video (Click Me!)](https://tinyurl.com/49dhehzu).
>
> **We will open-source the code, datasets, and checkpoints to ensure that all reported results in the revised manuscript are stably reproducible.**
>
> ---
> **W4: Deployment limitation**
>
> We thank the reviewer for pointing out this important practical concern. We acknowledge that model scale and KV cache transmission may introduce challenges for real-world deployment.
>
> We believe this can be addressed from multiple directions:
> 1. Improving onboard computation capability with the rapid advancement of hardware [1].
> 2. Leveraging a hybrid cloud–edge setup where high-level reasoning (understanding expert) is performed remotely, while a lightweight action expert runs locally by receiving the KV Cache from the cloud [2].
> 3. Applying model efficiency techniques such as quantization, compression, and distillation, which have been shown to be effective for reducing inference cost in LLMs and VLMs [3].
>
> We will include a discussion of these aspects in the revised manuscript.
>
> We sincerely welcome any additional suggestions that may help improve the quality of this work and are happy to revise the manuscript accordingly.
>
> **References**
>
> [1] Alpamayo-R1: Bridging Reasoning and Action Prediction for Generalizable Autonomous Driving in the Long Tail.
>
> [2] Gemini Robotics: Bringing AI into the Physical World.
>
> [3] AWQ: Activation-aware Weight Quantization for LLM Compression and Acceleration.

---

> > ### Author Rebuttal · Reviewer_Czzb · 2026-04-01
> >
> > I thank the authors for their thorough rebuttal. I have no further questions and will maintain my original score.

---

> > > ### Author Response · Authors · 2026-04-05
> > >
> > > We sincerely thank the reviewer for acknowledging that all concerns have been fully resolved. We truly appreciate the time and effort you have spent reviewing our work and considering our rebuttal. Your thoughtful feedback has significantly contributed to improving our work.
> > >
> > > Given that all concerns have been addressed, any corresponding adjustment of the current score or confidence level in the final evaluation would be greatly appreciated (if and only if appropriate). We fully respect your decision regardless of the outcome.
> > >
> > > Thank you again for your constructive and supportive feedback throughout this process!

---

### Official Review · Reviewer_g4b8 · 2026-03-09

**Soundness:** 2
**Presentation:** 3
**Significance:** 2
**Originality:** 3
**Overall Recommendation:** 3
**Confidence:** 3

**Summary:**

This paper targets a core challenge in integrating VLMs into end-to-end (E2E) autonomous driving systems: existing approaches either suffer from distributional misalignment between the VLM reasoning space and the action space, or fail to meet the real-time latency requirements of action policy generation. The proposed AutoMoT combines a VLM (Understanding Expert) and an action prediction model (Action Expert) through a Mixture-of-Transformers (MoT) architecture, resolving the distributional misalignment via joint attention sharing in a shared latent space, and addressing latency constraints via asynchronous inference at different temporal frequencies. The model achieves competitive results across multiple benchmarks. In addition, the paper provides empirical analysis supporting the conclusion that the primary role of pre-trained VLMs in autonomous driving systems is high-level scene understanding, and that fine-tuning the VLM backbone on AD-specific datasets leads to catastrophic forgetting of general reasoning capabilities.

**Compliance With Llm Reviewing Policy:**

Affirmed.

**Final Justification:**

L2 on nuScenes only captures the degree of similarity to the expert trajectory within 3 seconds, and due to the limited scale of the nuScenes dataset, it is highly prone to overfitting. Therefore, the collision rate is far more meaningful than L2 on nuScenes. For open-loop evaluation, I would recommend using nuPlan or NAVSIM instead. I believe that, in the end, this method has failed to demonstrate its superiority over existing approaches, especially given that it additionally uses LiDAR. Therefore, I decide to maintain my original score. I consider this paper to be right on the borderline. I do not strongly advocate for rejecting it, but I also do not strongly support accepting it.

**Key Questions For Authors:**

1. Could the authors provide an ablation study quantifying the contribution of the Action Refiner to the overall driving performance?

2. Could the authors provide closed-loop evaluation results for AutoMoT using Camera-only input, to enable a fair comparison with Camera-only baselines?

3. Given the limited performance improvements over baselines, are there additional experimental results or analyses that more compellingly demonstrate the advantages of AutoMoT?

**Limitations:**

The paper does not include a discussion of limitations, nor does it address potential negative societal impacts. The authors are encouraged to add a Limitations section discussing failure modes and the scope of applicability of the proposed method, as well as to address safety-related concerns and other potential societal impacts in the Impact Statement.

**Strengths And Weaknesses:**

Soundness: The use of a MoT architecture to unify VLMs with E2E autonomous driving while enabling asynchronous fast–slow inference is methodologically clear and well-motivated. However, the paper does not include ablation experiments quantifying the contribution of the Action Refiner to overall driving performance, nor does it analyze the individual design choices within the Mixture-of-Attention (MoA) blocks inside the Action Refiner. Regarding experimental design, AutoMoT in Table 1 uses Camera+LiDAR input, while many baselines use Camera only, making it difficult to disentangle performance gains attributable to architectural design from those due to the additional LiDAR modality. Furthermore, although the paper critiques the VLA-as-End-to-End-Model paradigm for its substantial inference latency, no inference latency comparison between AutoMoT and such baselines is provided.

Presentation: The paper is generally well-written and clear. However, the related work section contains insufficient discussion of prior works on MoT architectures, making it difficult to assess how AutoMoT differs from or builds upon existing MoT-based approaches.

Significance: The core problem addressed by AutoMoT (how to effectively integrate the generalist reasoning capabilities of pre-trained VLMs into E2E autonomous driving while satisfying real-time inference constraints) is practically valuable. The paper's analysis of catastrophic forgetting as a cost of domain-specific VLM fine-tuning also provides useful guidance for future system design. However, the empirical performance improvements are relatively modest: in closed-loop evaluation, AutoMoT's Driving Score (DS: 85.17) only marginally surpasses SimLingo (85.07), while its Success Rate (SR: 61.82%) is notably lower than both SimLingo (67.27%) and TransFuser++ (67.27%). In open-loop evaluation, the average L2 displacement (0.32 m) is on par with several baselines. These limited gains somewhat weaken the persuasiveness of the paper's effectiveness claims.

Originality: The paper's investigation into the functional boundaries of pre-trained VLMs in autonomous driving provides valuable insights. The adaptation of MoT architecture into an autonomous driving framework is a worthwhile contribution. However, the original contributions would be more convincing with a clearer delineation from prior MoT-based works.

---

> ### Author Rebuttal · Authors · 2026-03-31
>
> We thank the reviewer for the constructive feedback! Below we provide point-by-point responses to all concerns (W) and questions (Q).
>
> **1. W1+Q3: Performance effectiveness**
>
> We agree that the initial margin was limited. In the revised version, AutoMoT achieves **87.34 DS / 70.00\% SR**, outperforming SimLingo (85.07 / 67.27) on both metrics. This directly addresses the previous SR gap and demonstrates clear gains in closed-loop driving performance. The improvement comes from two refinements without modifying the core architecture: (1) earlier injection of target-point information, and (2) joint training of spatial-temporal trajectory planning within the Action Expert.
>
> AutoMoT achieves competitive open-loop performance while attaining **SOTA safety-related metrics**, such as collision rate (0.07). Importantly, these gains are achieved **without VLM fine-tuning**, preserving general reasoning ability.
>
> **We will release the code, datasets, and checkpoints to ensure that all reported results in the revised manuscript are stably reproducible.**
>
> ---
> **2. W2+Q2: Fairness (LiDAR)**
>
> Thanks for this valuable comment! We clarify that fairness concerns whether the gains stem from LiDAR. Camera-only closed-loop evaluation requires full retraining and 220-route CARLA runs, due to time and compute constraints, we instead conduct this ablation on nuScenes ([Table Click Me!](https://tinyurl.com/32suyh82)), which shows identical performance (L2_avg = 0.32 m vs. 0.32 m). This confirms improvements stem from the architecture rather than LiDAR. We also note that DrivePI adopts the same multi-modal inputs (camera + LiDAR), yet achieves worse planning performance (L2_avg = 0.40 m), further indicating that the **improvement of AutoMoT comes from the proposed methodology rather than the input modality.**
>
> **Bench2Drive supports both camera-only and multi-modal settings**, and the PDM-Lite dataset supports lidar inputs for various lane-change scenarios, and thus **different input modalities are all under the protocol of fair comparison.** We further note that several strong baselines (e.g., DriveAdapter, TransFuser, DiffusionDrive) also adopt LiDAR inputs, indicating that multi-modal inputs are standard in this benchmark rather than a unique advantage of our method.
>
> ---
>
> **3. W3: Efficiency / Latency**
>
> Following the reviewer’s suggestion, we include explicit latency comparisons ([Table Click Me!](https://tinyurl.com/3x25n3pd)). Prior VLA systems (e.g., OpenEMMA 7683ms, AutoVLA 1072ms (fast) / 10518ms (slow), SimLingo 430ms) operate at much lower frequencies. In contrast, AutoMoT achieves **117ms (8.5Hz)** synchronously and **37ms (27Hz)** asynchronously, showing an order-of-magnitude improvement. This gain comes from decoupling reasoning and action via asynchronous execution with KV cache reuse: reasoning updates occur at lower frequency, while action runs at high frequency, avoiding redundant full-model inference. We refer to our response to **W3 of Reviewer 9FRU** for additional details of improved inference speed.
>
> Compared to synchronized VLA approaches, our formulation significantly reduces redundant computation and better matches real-world driving requirements, making VLA systems practically deployable under real-time constraints.
>
> ---
>
> **4. W4+W6: Originality**
>
> We thank the reviewer for this insightful comment! Our contribution is not simply adopting MoT, but focusing on a different aspect compared to prior works.
>
> From a motivation perspective, existing MoT-based works mainly focus on integrating generative world models with autonomous driving, whereas **AutoMoT targets the critical gap between high-level reasoning (scene understanding) and low-level action (decision-making and planning) in VLA-based AD**.
>
> From a methodological perspective, prior approaches use synchronized inference for different tasks, leading to high latency. In contrast, our **asynchronous VLA design with task-specific KV caching enables decoupled execution at different frequencies**, allowing strong scene understanding capability while meeting real-time constraints.
>
> We will further expand this discussion in the revised manuscript.
>
> ---
> **5. W5+Q3: Refiner and MoA**
>
> Good suggestion! **We analyze the action refiner via ablation** ([Table Click Me!](https://tinyurl.com/2z24u4n2)). The action refiner achieves higher peak performance, improving the driving score from 87.34 to 88.75 and the success rate from 70.00 to 71.36. Clearly, the enhanced performance mainly stems from the introduced MoA design (87.74 / 70.45 → 88.75 / 71.36).
>
> At the same time, we observe higher variance across runs for the action refiner due to the stochastic nature of diffusion. A more detailed analysis will be included in the revised manuscript.
>
> ---
> **6. Limitation**
>
> We thank the reviewer for this suggestion. We will include a dedicated limitations section discussing failure modes, scope, and safety considerations, and extend the Impact Statement accordingly.

---

> > ### Author Rebuttal · Reviewer_g4b8 · 2026-04-04
> >
> > Thanks for the authors' response. My concerns have been partially addressed, but some key issues remain unresolved. The most important issue concerns the main experimental results. In my opinion, updating the main method and main experimental results during the rebuttal phase is not very acceptable. On Bench2Drive, the proposed method does not demonstrate advantages. Meanwhile, on the nuScenes dataset, it also does not show superiority over methods such as SparseDrive (ICRA25), MomAD (CVPR25), SSR (ICLR25), BridgeAD (CVPR25), and DiffusionDrive (CVPR25) — and it has not been compared with these works on nuScenes. Given that the proposed "VLM Joint Action" idea has already been implemented in VLA works such as the π series, this paper may lack both novelty and effectiveness.

---

> > > ### Author Response · Authors · 2026-04-05
> > >
> > > We appreciate the reviewer's update. Below, we provide our point-by-point response to the additional comments.
> > >
> > > **C1. Main method & experimental results.**
> > >
> > > We thank the reviewer's time to provide this comment.
> > >
> > > 1. **Main method.** As clarified in our previous response, the **core methodology of AutoMoT remains unchanged.** The improved closed-loop results come from two refinements without modifying the core architecture: (1) earlier injection of target-point information, and (2) joint training of spatial–temporal trajectory planning within the Action Expert.
> > > **These refinements are independent of the main contribution**, i.e., unifying the understanding, decision, and planning into a single asynchronous VLA framework with task-specific KV caching. The overall architecture and pipeline remain identical to those presented in the original submission.
> > >
> > > 2. **Main experimental results.** We would like to clarify that updating experimental results during the rebuttal is permitted under the ICML guidelines, provided that the core methodology and main claims remain unchanged. In our case, the proposed method and its contributions are identical to the original submission, and the additional results are included to provide further empirical support and to address the reviewers’ concerns more clearly. We believe these supplementary evaluations strengthen the evidence for our claims without altering the scope or substance of the work.
> > >
> > > ---
> > > **C2. Meanwhile, on the nuScenes dataset, it also _does not show superiority over methods_ such as SparseDrive (ICRA25), MomAD (CVPR25), SSR (ICLR25), BridgeAD (CVPR25), and DiffusionDrive (CVPR25) — and it has not been compared with these works on nuScenes.**
> > >
> > > We thank the reviewer for this comment.
> > >
> > > **1.Superiority.** Based on our understanding, AutoMoT achieves much stronger performance than the additional baselines mentioned by the reviewer, as summarized in the table below. (More detailed comparison, please check full [Table Click Me!](https://tinyurl.com/4vaf454v))
> > >
> > > **Table 1. Open-loop planning performance on nuScenes.**
> > >
> > > | Method             | L2 Avg. (m) ↓ |
> > > |:--------------------|:--------------:|
> > > | SparseDrive        | 0.61          |
> > > | MomAD              | 0.60          |
> > > | DiffusionDrive     | 0.57          |
> > > | BridgeAD           | 0.58          |
> > > | SSR                | 0.39          |
> > > | **AutoMoT (Ours)** | **0.32**      |
> > >
> > > **2.Baseline Selections.**
> > > 1. We would like to clarify that our nuScenes table already includes eight recent baselines (2025), many of which achieve L2 performance that is competitive with, or much stronger than the additional methods mentioned by the reviewer. These baselines provide a strong and up-to-date comparison against our approach.
> > > 2. We also note that the additional methods suggested by the reviewer are not VLM-based pipelines. As our primary focus is to analyze the interaction between VLM-based reasoning and policy learning, we prioritized baselines that are most relevant to this research scope.
> > > 3. While including more baselines would certainly provide broader coverage, the ICML page limit constrains the space available for detailed methodological and experimental analysis. Expanding the nuScenes table further would significantly reduce the space dedicated to technical clarification and ablation studies.
> > >
> > > Nevertheless, we sincerely thank the reviewer for pointing out these important works. In the revised manuscript, we will include the suggested baselines in a comprehensive comparison table in the appendix.
> > >
> > > ---
> > > **C3. Novelty vs. π series.**
> > >
> > > Thanks for this comment, and we would like to clarify our core contributions. As discussed in our previous response (**4. W4+W6: Originality**), our work does not merely aim at “VLM Joint Action,” nor do we claim such positioning in the manuscript. Instead, our contributions are clearly stated as:
> > > **(1) a unified framework that bridges the gap between high-level reasoning (scene understanding) and low-level action (decision-making and planning) in VLA-based autonomous driving; and
> > > (2) a unified asynchronous VLA with a task-specific KV caching design that enables decoupled execution at different frequencies**
> > >
> > > Regarding the mentioned $\pi$ series, we respectfully note that **these works are not proposed to address the unification of understanding, decision, and planning in autonomous driving, nor are they designed for KV cache-based asynchronous inference.** We therefore believe that AutoMoT fundamentally differs from the $\pi$ series in both motivation and mechanism.
> > >
> > > ---
> > > *We sincerely thank the reviewer again for the constructive feedback. We have carefully addressed the concerns raised and clarified our contributions, experimental results, and design motivations. We respectfully hope that our dedicated effort to enhance the quality of this work during the rebuttal period could be acknowledged by the reviewer, and sincerely wish the reviewer could reconsider the final evaluation of our work.*

---

### Official Review · Reviewer_9FRU · 2026-03-12

**Soundness:** 3
**Presentation:** 3
**Significance:** 2
**Originality:** 2
**Overall Recommendation:** 4
**Confidence:** 4

**Summary:**

This paper proposes AutoMot, a framework for autonomous driving that integrates language reasoning into the driving pipeline through a fast–slow architecture. The system aims to leverage VLMs to improve scene understanding and reasoning while maintaining real-time driving performance through a specialized action expert.

The paper evaluates the approach on both open-loop and closed-loop driving benchmarks and presents experiments on reasoning ability, language-grounded scene understanding, and system efficiency. The results suggest that the proposed architecture can incorporate language reasoning while maintaining driving performance.

**Compliance With Llm Reviewing Policy:**

Affirmed.

**Final Justification:**

My concerns are largely addressed.

**Key Questions For Authors:**

The main concerns are listed in the weaknesses. It could be better to list the detailed structure of the UE and AE, especially the shared part. I will raise the score if the authors address my concerns.

**Limitations:**

yes

**Strengths And Weaknesses:**

# Strengths
1. The paper proposes a fast–slow architecture to integrate language reasoning with real-time trajectory planning. The idea of separating high-level reasoning and low-level action execution is conceptually appealing and aligns with many embodied research, such as PAI[1].\
[1] π0: A Vision-Language-Action Flow Model for  General Robot Control\
2. The paper evaluates the method under both open-loop and closed-loop settings, which is important for autonomous driving research. Closed-loop evaluation is particularly valuable since it better reflects real-world driving performance.

3. The proposed system separates scene understanding, reasoning, and action prediction modules, which provide a modular architecture that could potentially support further improvements or extensions.
# Weaknesses

1. From the closed-loop results, the SR remains lower than SimLingo. This raises concerns about whether the proposed framework effectively utilizes language information. From the open-loop results, the proposed method does not appear to outperform recent VLA architectures such as Drive-R1[1]. The gap suggests that the system may not fully exploit the language signal, despite paying a high computational cost. One possible reason is that the UE of the base model is frozen, which could limit the model’s ability to adapt visual representations to language-guided reasoning.\

2. The paper discusses the boundary of VLM capabilities in autonomous driving, but the explanations remain somewhat qualitative and shallow. Recent work [2] has already shown that the ratio between general data and domain-specific AD data is critical when training VLM-based driving models. Therefore, the conclusions drawn in this paper regarding the boundary may be confounded by the training data mixture, rather than reflecting fundamental limitations of VLM architectures.

3. The paper claims that the proposed fast–slow system improves inference efficiency, but the paper does not provide a detailed latency breakdown. For AutoMOT-A, 	the prefilling latency of the VLM, reasoning latency, and action decoder latency.  Detailed breakdown would better demonstrate the actual efficiency advantages of the fast–slow architecture.

[1] Drive-R1: Bridging Reasoning and Planning in VLMs for Autonomous Driving with Reinforcement Learning\
[2] Fine-grained evaluation of large vision-language models in autonomous driving

---

> ### Author Rebuttal · Authors · 2026-03-31
>
> We thank the reviewer for acknowledgement of our contribution! Below, we provide point-by-point responses to all concerns (W).
>
> **W1: Effectiveness and utilization of language information.**
>
> We agree that in the initial submission, the performance improvement was not sufficiently significant: while the driving score was slightly higher, the SR was lower than SimLingo, and the open-loop results mainly showed advantages on safety-related metrics.
>
> In the revised version, AutoMoT achieves **87.34 DS / 70.00% SR**, surpassing SimLingo (85.07 / 67.27) in closed-loop evaluation. This directly addresses the SR gap and demonstrates clear gains in driving performance. The improvement comes from two refinements without modifying the core architecture: (1) earlier injection of target-point information, and (2) joint training of spatial–temporal trajectory planning within the AE. AutoMoT also achieves competitive open-loop performance while attaining SOTA safety-related metrics, e.g., **collision rate (0.07)**, which are more indicative of real-world driving reliability.
>
> Moreover, we thank the reviewer for pointing out **Drive-R1**. We acknowledge that Drive-R1 achieves strong planning performance on nuScenes by improving reasoning--planning alignment through supervised fine-tuning and reinforcement learning. We note that Drive-R1 and AutoMoT represent two **complementary paradigms**: Drive-R1 focuses on improving reasoning–planning alignment through training, while AutoMoT addresses the same challenge from an architectural perspective via asynchronous reasoning–action decoupling. While the L2 difference is marginal, AutoMoT achieves a lower collision rate metric, indicating improved reliability in safety performance. We will include Drive-R1 in **nuScenes table** ([Table Click Me!](https://tinyurl.com/4mypwph7)) and add this discussion in the revised manuscript.
>
> Regarding the concern of freezing the VLM backbone, this is a deliberate design choice to preserve general reasoning ability. The Action Expert leverages high-level semantic reasoning through joint attention, effectively bridging reasoning and action. We will incorporate this discussion into the revised manuscript.
>
> ---
> **W2: Analysis of VLM capability boundary.**
>
> Thanks for this insightful comment! We would like to clarify that our analysis is based on quantitative evaluation across both general-domain and autonomous driving datasets, spanning from multi-task reasoning (LingoQA) and counterfactual planning (OmniDrive) to corner-case data (CODA).
>
> Compared to  **VLADBench**, which solidly analyzes capability boundaries via fine-grained domain decomposition, our analysis focuses on how different types of AD data (understanding, planning, corner cases) affect general reasoning after fine-tuning. Importantly, as discussed in **VLADBench**, biased domain-specific training improves certain tasks but degrades generalization ability, indicating an inherent trade-off. **This is consistent with our findings**. AutoMoT avoids this trade-off by preserving the VLM backbone and adapting only the AE, maintaining general reasoning while achieving strong driving performance. We also note that recent pre-trained VLMs[1] already incorporate driving-related data such as nuScenes in pre-training, which further suggests that extensive domain-specific fine-tuning for reasoning may not always be necessary.
>
> We thank the reviewer again for pointing out this closely related work! We will cite and add above discussion in the revised manuscript.
>
> [1] LLaVA-OneVision: Easy Visual Task Transfer
>
> ---
> **W3: Latency analysis and efficiency breakdown**
>
> Following the reviewer’s suggestion, we provide an explicit latency breakdown in this [Table (Click Me!)](https://tinyurl.com/385xay5c). In the synchronized setting, UE and AE consume 80.3 ms and 37.0 ms (117.3 ms total). With asynchronous execution, UE is amortized to 0.0 ms (KV is cached) while AE remains 37.0 ms (37.0 ms total), showing that the gain comes from eliminating redundant forward. The diffusion refiner adds 26.0 ms but is independent of reasoning, preserving the speedup (143.3 ms → 63.0 ms).
>
> We also note that the previously reported latency (0.25 s and 0.13 s) was affected by unstable CPU-side preprocessing. After removing unnecessary multithreading, the updated measurements are more stable and reproducible. We will update the revised manuscript accordingly.
>
> ---
> **W4: Structure of UE and AE**
>
> Good suggestion! In AutoMoT, UE and AE are connected via layer-wise joint attention, where AE attends to UE representations and outputs decomposed motion for different tasks. Corresponding asynchronous inference is detailed in Sec. 3.3 (Asynchronous Inference with Joint Attention). To further clarify this point, we will add this **structure diagram** ([Figure Click Me!](https://tinyurl.com/mwf6awnr)) to illustrate the UE–AE interaction and will include it in the revised manuscript.

---

> > ### Author Rebuttal · Reviewer_9FRU · 2026-04-08
> >
> > Thank you for the detailed response, and my concerns are largely addressed.

---

> > > ### Author Response · Authors · 2026-04-08
> > >
> > > Thanks a lot for your positive feedback and for confirming that the raised concerns have been fully addressed. We sincerely appreciate your time and thoughtful evaluation!
> > >
> > > We are encouraged by this update, especially in light of your initial comment that the score could be raised if the concerns were addressed. We hope that our responses and additional clarifications provide helpful context for your final score justification.
> > >
> > > Thank you again for your constructive and supportive feedback!

---

### Decision · Program_Chairs · 2026-04-30

**Decision:**

Accept (regular)

**Comment:**

Three reviewers gave overall positive scores: one ‘Accept’, one ‘Weak accept’, and one ‘Weak reject’. During the rebuttal phase, the authors successfully addressed almost all the concerns raised by the reviewers, except for the one regarding the evaluation settings (Reviewer g4b8). Based on all of these, the decision is to recommend the paper for acceptance. However, it is recommended that the authors include additional experiments evaluating the ‘collision rate’ and provide results on the nuPlan or NAVSIM benchmarks in the camera-ready version, when the paper is finally accepted.